# Research on Speed Control Strategies for Explosion-Proof Diesel Engine Monorail Cranes

**Hai Jiang [1], Dongjie Wang [1], Jiameng Cheng [1], Penghui Li [2], Xiaodong Ji [3,\*], Yang Shen [4] and Miao Wu [1]**

[1] School of Mechanical Electronic and Information Engineering, China University of Mining & Technology, Beijing 100083, China; bqt1900401002@student.cumtb.edu.cn (H.J.); wdj@student.cumtb.edu.cn (D.W.); cjm@student.cumtb.edu.cn (J.C.); wum@cumtb.edu.cn (M.W.)

[2] CCTEG Taiyuan Research Institute Co., Ltd., Taiyuan 030032, China; 15303417101@163.com

[3] State Key Laboratory of Mechanical Behavior and System Safety of Traffic Engineering Structures, Shijiazhuang Tiedao University, Shijiazhuang 050043, China

[4] School of Vehicle and Transportation Engineering, Tsinghua University, Beijing 100084, China; shenyang@mail.tsinghua.edu.cn

\* Correspondence: jxd2022@stdu.edu.cn; Tel.: +86-173-0332-6056

**Abstract:** This paper introduces a control method tailored for the speed regulation of monorail cranes in coal mines. Initially, an analysis of the structure and load conditions of the monorail crane drive components is conducted to calculate the traction force, clamping force, and target travel speed across varying operational scenarios. Subsequently, the hydraulic system schematic of the monorail crane is analyzed to develop a mathematical model for speed control, enabling the assessment of system stability using transfer functions. A simulation model of the monorail crane speed control loop is then created in AMESim, where fuzzy adaptive PID controllers and MPC controllers are optimized in a collaborative simulation with Simulink. Experimental findings reveal that in a single acceleration condition, both controllers demonstrate superior dynamic response compared to a traditional PID controller, with the MPC controller exhibiting an overshoot of merely 8.9%. In speed variation conditions, the MPC controller achieved a settling time in the range of 0.26–0.3 s. Notably, the MPC controller displays a maximum overshoot of 11%, substantially enhancing the dynamic response performance of speed regulation in monorail cranes.

**Keywords:** monorail crane; hydraulic system; speed control loop; simulation model; fuzzy adaptive control; MPC controller





## 1. Introduction

Coal is the most economical and reliable resource among primary energy sources in China [1]. Traditional coal production processes involve tunneling, mining, and transportation. As mining capabilities have advanced, there is an increased demand for efficient transportation [2]. Coal mine transportation is categorized into main and auxiliary types: the former handles raw coal transport, while the latter manages the transport of materials, equipment, and personnel [3]. Auxiliary transportation equipment can be classified into two main types based on their driving mechanism: rail-guided locomotives and trackless vehicles. Rail-guided locomotives offer several advantages over trackless vehicles, including greater traction, superior climbing ability, lower energy consumption, and reduced maintenance costs. These attributes make them well suited for transporting spoil, materials, and personnel over long distances and steep gradients in tunnels [4]. Diesel-powered monorail cranes are the primary equipment used for rail-guided auxiliary transportation in coal mines.

Monorail cranes primarily operate under manual control. The track of the monorail crane undulates according to the ceiling profile, and the route includes wind doors, switches, and gradients that necessitate real-time adjustments to speed and driving force during

operation. Consequently, driving strategies designed for ground-based rail vehicles that maintain constant speed or power are not applicable to monorail cranes functioning in complex environments. The speed control of a monorail crane must consider factors such as load, operating speed, and fluctuations in working conditions to ensure both stability and safety during operation, as well as the effectiveness of the control strategy. Achieving efficient and safe autonomous driving is contingent upon adaptive optimization based on operational conditions and real-time power matching under load-sensitive contexts.

The foundation for achieving speed regulation in monorail cranes lies in the in-depth study of their hydraulic drive system, which is a closed hydraulic transmission system. Quan Fei [5] studied the drive system of electric traction monorail cranes, exploring the relationship between crane working parameters and the system's dynamic characteristics using a 3D model and dynamic simulations. Chen Yu [6] developed a 3D model of the crane's drive unit, analyzed the principles of the drive, clamping, and braking hydraulic systems, and built related mathematical models to analyze dynamic performance. He also used AMESim to model and simulate the key hydraulic systems of the drive unit and carried out optimization design. Wang Xu [7] designed a distributed hybrid power system architecture for diesel-powered monorail cranes, created a physical model of the hybrid power system using AMESim, and investigated the feasibility and energy-saving performance of various control strategies based on actual working conditions. Li Kun-quan [8] established a 3D model of the driving mechanism with SolidWorks, conducted force analysis, created a simplified hydraulic circuit with AMESim, and obtained curves for motor speed, output torque, and pump oil pressure variations based on working conditions. Swider, J. et al. [9] proposed a two-stage braking algorithm and analyzed the effects of changing braking algorithm parameters on deceleration, braking time, and braking distance through numerical simulations. The new algorithm can further increase monorail crane operating speed while ensuring safe braking. These studies primarily focus on the dynamic models and force states of monorail crane drive systems, but research on optimizing their dynamic characteristics, particularly adaptive speed optimization matching crane motion states, is limited.

With the development of auxiliary transportation in coal mines, integration research of perception technology and control technology has attracted widespread attention [10]. Multi-modal perception technology is the basis for the autonomous judgment of the operating state of monorail cranes. Jiang, H. et al. [11] proposed an error compensation method based on the dynamic model of the monorail crane driver's cab unit and a multi-subsample error compensation algorithm, achieving high-precision perception of the motion posture and heading of the monorail crane through an inertial navigation system. Li Honggang et al. [12], based on the fusion of laser radar and millimeter wave radar in a multi-target recognition architecture, applied the joint probabilistic data association (JPDA) algorithm based on Kalman filtering on the basis of data correlation to achieve multi-target recognition in the mining area environment and built a mine car unmanned driving system platform. Liu, Z. C. et al. [13] proposed a dynamic inclination estimation method for the monorail crane by using the Estimate-FocusedEKFNet algorithm to estimate the running inclination of the monorail crane based on the real-time collection of acceleration, running speed, and motor output torque parameters by sensors. The above research on motion posture and motion state perception of monorail movement provides the basis for the formulation of an adaptive speed control strategy for a monorail crane based on its motion state in this paper.

In this paper, the structure of the monorail crane drive unit, its motion state, and load characteristics are combined to establish a mathematical model of the speed control loop, obtain the transfer function of the monorail crane speed control process, and conduct stability analysis. Furthermore, based on the mathematical model and hydraulic schematic, an AMESim simulation model of the monorail crane speed control loop is established to match the parameters of different operating conditions with the speed control process of the monorail crane. By using a combined simulation interface, different speed control strategy algorithms are incorporated into the simulation model to reveal the response

characteristics of the monorail crane speed control loop under different algorithms and to derive the optimization results for each algorithm.

## 2. Analysis of the Monorail Crane Drive Unit Structure and Load Characteristics

### 2.1. Drive Unit Structure

The DC280/160Y explosion-proof diesel engine monorail crane's driving and braking actions are typically managed by $N$ ($N = 2 \sim 10$) drive units, each comprising a drive unit, a clamping unit, and a braking unit [14]. The operating conditions in the tunnel are relatively complex, with slopes generally not exceeding 25° and loads ranging from 15 to 65 tons. The running speed must be adjusted according to the slope and load conditions. Therefore, monorail cranes are usually equipped with a coasting function. When operating without a load, some drives can be disengaged to increase speed, while under heavy load, all drives are engaged to ensure adequate traction [15]. The drive force of the explosion-proof diesel engine monorail crane investigated in this paper can reach up to 28 kN, and its three-dimensional model is illustrated in Figure 1.

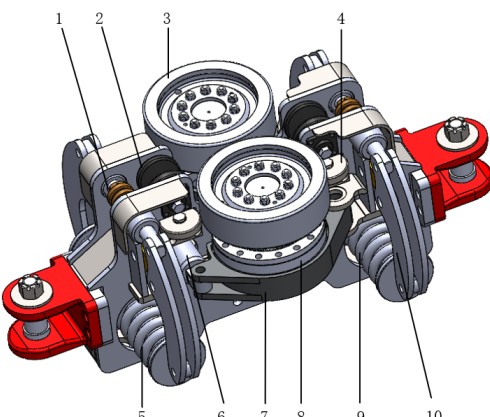

**Figure 1.** Three-dimensional model of monorail crane drive unit. 1—Brake shoe; 2—guide wheel; 3—friction drive wheel; 4—guide wheel; 5—brake spring; 6—clamping oil cylinder; 7—motor housing; 8—drive motor; 9—brake oil cylinder; 10—brake arm.

### 2.2. Force and Load Characteristics of the Drive Unit

The force dynamics of the monorail crane drive unit during actual operation are complex. The slippage of the friction drive wheel under operating and track conditions impacts the force exerted on the drive unit. This analysis assumes that the friction drive wheel does not experience slippage or deformation during movement, simplifying the force dynamics of the drive unit during operation, as illustrated in Figure 2.

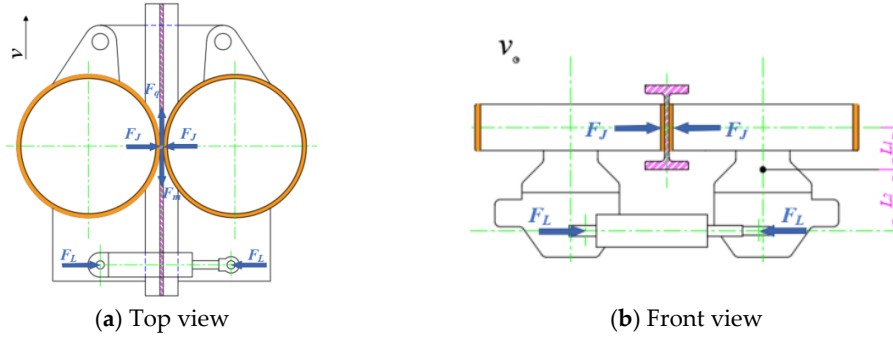

| (**a**) Top view | (**b**) Front view |
|:---:|:---:|

**Figure 2.** Schematic of the forces acting on the friction drive wheel of the drive unit.

The traction force is represented by $F_q$, while $F_m$ denotes the frictional force acting on the drive friction wheel in the direction of travel. $F_J$ is the clamping force exerted

on the contact point between the friction drive wheel and the track by the clamping oil cylinder, and $F_L$ refers to the clamping force applied on the motor housing by the clamping oil cylinder. The vertical restriction of the drive unit is supported on the track surface by steel guide wheels, offering high stiffness and minimal deformation. To simplify the analysis, the friction between the guide wheel and the track, as well as the vertical displacement and friction of the friction drive wheel, are neglected during horizontal motion. By determining the position of the vertical center of gravity from the front view, a set of force equilibrium equation, Equation (1), can be derived to establish the balanced state during horizontal movement.

$$\begin{cases} F_J L_1 = F_L L_2 \\ F_q/2 = \mu F_J \end{cases} \tag{1}$$

When the monorail crane operates in an inclined tunnel, the force of the load on the craning mechanism generates a drag force component in the direction of travel on the drive unit. Disregarding any deformation during travel, the force state of the monorail crane is depicted in Equation (2).

$$\begin{cases} F_q = F_g + F_a + F_f + F_{f_z} \\ F_g = (G_q + G_d) \sin\alpha \\ F_a = (m_q + m_d) \gamma a \\ F_f = \mu(m_q + m_d) \cos\alpha \\ F_{f_z} = 2 f_z F_m i_q \end{cases} \tag{2}$$

where $F_g$ represents the gravitational component of the entire vehicle's weight in the direction of the slope, kN; $G_q$ indicates the self-weight of the drive unit, kN; $G_d$ represents the total weight of the lifting load, kN; $\alpha$ signifies the slope of operation in degrees; $\gamma$ denotes the inertia coefficient, generally taken as 1.075 for mining equipment; $a$ indicates the acceleration of the monorail crane during operation. Under normal conditions, when heavily loaded, $a = 0.015 \text{ m/s}^2$, and when unloaded, $a = 0.3 \text{ m/s}^2$, where $\mu$ represents the coefficient of friction between the drive wheel and steel rail, $f_z$ signifies the resistance coefficient, $F_m$ is the hydraulic motor transmission resistance, and $i_q$ indicates the number of drive units put into operation.

The equations described above provide the calculation formula for the traction force of the monorail crane under slope operating conditions, as depicted in Equation (3).

$$F_{uq} = (m_q + m_d)(g\sin\alpha + \gamma a + \mu g \cos\alpha) + 2 f_z F_m i_q \tag{3}$$

The calculation from equation set (1) infers that the clamping force required for the friction drive wheel in this state is as follows:

$$F_{uJ} = \frac{(m_q + m_d)(g\sin\alpha + \gamma a + \mu g \cos\alpha) + 2 f_z F_m i_q}{2\mu} \tag{4}$$

The clamping cylinder needs to provide a clamping force as follows:

$$F_L = \frac{F_J L_1}{L_2} \tag{5}$$

Similarly, when operating under downhill conditions, the monorail crane's required traction force is determined by Equation (6):

$$F_{dq} = (m_q + m_d)(\mu g \cos\alpha - g\sin\alpha - \gamma a) + 2 f_z F_m i_q \tag{6}$$

### 2.3. Analysis of Operating Conditions for Monorail Cranes

During monorail crane operation in underground tunnels, its conditions and speeds are categorized based on factors such as load capacity and tunnel gradient. Ignoring the

frictional resistance of the drive unit wheels and using relevant technical parameters, the average maximum haulage weight per drive unit can be calculated with Equation (3). This calculation is based on a monorail crane with a self-weight of 15 tons and a maximum traction force of 280 kN for ten drive units.

The monorail crane's transportation capacity curve indicates that when the tunnel gradient is below 5°, the crane's maximum transportation speed is 2.0 km/h, both when unloaded and fully loaded. The maximum permissible gradient for operation should not exceed 25°, as specified by the Coal Mine Safety Regulations. To facilitate future research and based on the driving habits of monorail crane operators under varying conditions, the operational states of the monorail crane are roughly categorized as illustrated in Table 1.

**Table 1.** Vehicle speeds of the monorail crane under different operating conditions.

| Total Weight | Gradient | Speed |
|:---:|:---:|:---:|
| $M \leq 30$ t | $\alpha \leq 5°$ | 2.0 m/s |
| 30 t $\leq M \leq 60$ t | $\alpha \leq 5°$ | 2.0 m/s |
| $M \leq 30$ t | $5° \leq \alpha \leq 15°$ | 2.0 m/s |
| 30 t $\leq M \leq 60$ t | $5° \leq \alpha \leq 15°$ | 0.9$\sim$1.5 m/s |
| $M \leq 30$ t | $15° \leq \alpha \leq 25°$ | 0.5$\sim$1.0 m/s |
| 30 t $\leq M \leq 60$ t | $15° \leq \alpha \leq 25°$ | 0.5 m/s |

## 3. Mathematical Model and Control Principles of Speed Control System

The monorail crane is powered by a diesel engine, which drives the main pump and gear pump to convert mechanical energy into hydraulic energy. The hydraulic system utilizes a closed variable system to control the movement functions, including running, stopping, and brake operation. The motor speed and output efficiency are determined by the pump displacement. This system is crucial for regulating the speed of the monorail crane [16]. The brake system and the clamping system both serve the speed control system. Owing to the complexity of the monorail crane system and the constraints of real vehicle testing conditions, this paper will employ simulation models for validation to examine control strategies for the movement of the monorail crane.

### 3.1. Hydraulic Drive System

The drive unit's main pump employs a closed-loop hydraulic control axial piston variable pump with an integrated slippage oil circuit. Each drive unit features a pair of drive motors arranged symmetrically on the left and right. The oil circuits for the drive motors on each side are connected in parallel, and the motors within the same unit rotate in opposite directions. This configuration primarily facilitates both driving and speed regulation functions during operation.

The schematic diagram of the hydraulic speed regulation system is presented in Figure 3. Upon starting the monorail crane, the diesel engine drives the hydraulic main pump, which delivers high-pressure oil through the filter, proportional valve oil block, power valve, flushing valve, and cut-off hydraulic control valve to the bidirectional drive motor. The output oil circuit is controlled to alter the motor's rotational direction, enabling the monorail crane to move forward or backward along the track. The motors in each drive unit are connected in parallel, leveraging the self-balancing characteristics of the parallel hydraulic system to ensure that the driving forces remain consistent, thereby guaranteeing synchronized movement of the drive units [17]. To simplify the system representation, one constant drive unit and one cut-off drive unit are utilized to depict the hydraulic motor section.

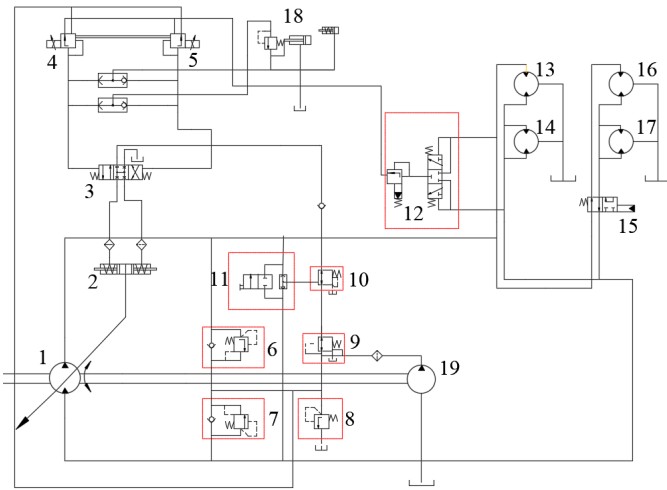

**Figure 3.** Schematic diagram of the monorail crane speed regulation circuit. 1—Variable pump; 2—variable cylinder; 3—hydraulic control main valve; 4, 5—proportional pressure reducing valve; 6, 7—high-pressure safety valve; 8—relief valve; 9—pressure reducing valve; 10—pressure reducing relief valve; 11—manual reversing valve; 12—flushing valve group; 13, 14—constant drive motor; 15—reversing valve; 16, 17—cut-off drive motor; 18—power valve group.

### 3.2. Analysis of the Speed Regulation Characteristics of the Drive System

According to the hydraulic principles of the monorail crane drive system, the entire system is a volumetric speed control circuit of a "variable pump + fixed displacement motor". Assuming the volumetric efficiency of the circuit is not considered, its speed–load characteristics are as follows:

$$n_M = \frac{V_B n_B}{V_M} \tag{7}$$

The displacement of the fixed displacement motor ($V_M$) is constant. When the speed of the variable pump ($n_B$) remains constant, the motor speed ($n_M$) varies proportionally with changes in the variable pump displacement ($V_B$). Consequently, the proportional direction control valve in the speed control circuit can be regulated via an electrical signal, thereby affecting the operation of the variable pump's control cylinder.

The expression for the thrust of the proportional solenoid can be determined by the current ($I$) flowing through the coil and the displacement ($x_T$) of the armature.

$$F_C = K_I I - K_l x_T \tag{8}$$

where $F_C$ is the thrust output of the proportional solenoid, N; $K_I$ is the current gain of the proportional solenoid, $N/A$, with $K_I = \frac{\partial F_C}{\partial I}$; $K_l$ is the displacement force increment added to the spring stiffness.

The force equilibrium equation for the electromagnetic assembly is the following:

$$F_C = m_T \frac{\partial^2 x_T}{\partial t^2} + C_T \frac{\partial x_T}{\partial t} + T \tag{9}$$

where $m_T$ is the mass of the armature, kg; $C_T$ is the damping coefficient; $T$ is the external load.

Equations (8) and (9) describe the dynamic behavior of the proportional solenoid as a function of the input current. By applying the Laplace transform and rearranging the resulting expressions, we obtain the following:

$$I(s) = \frac{m_T x_T(s)s^2 + C_T x_T(s)s + K_l x_T(s) + T}{K_I} \tag{10}$$

For the directional valve spool, the force exerted on it is equal in magnitude but opposite in direction to the external load force. Thus, the relationship is given by $F_x = T$ & $x_v = x_T$. The dynamic equilibrium equation for the spool is the following:

$$F_x = m_v \frac{d^2 x_v}{dt^2} + B_v \frac{dx_v}{dt} + K_v x_v + K_{fv} x_v \tag{11}$$

where $m_v$ is the mass of the spool, kg; $B_v$ is the viscous damping coefficient of the spool, $\text{N} \cdot \text{s/m}$; $K_v$ is the spring stiffness of the centering mechanism for the spool, N/m; $K_{fv}$ is the steady-state fluid dynamic stiffness coefficient acting on the spool, N/m.

By applying the Laplace transform to Equation (11) and integrating it with Equation (8) through (10), we obtain Equation (12):

$$\frac{x_v(s)}{I(s)} = \frac{K_I}{(m_T + m_v)s^2 + (C_T + B_v)s + \left(K_l + K_v + K_{fv}\right)} \tag{12}$$

The simplified transfer function of the input current to the spool displacement for the proportional directional control valve is obtained.

$$\frac{x_v(s)}{I(s)} = \frac{K_{bv}}{\frac{s^2}{\omega_{bv}^2} + \frac{2\xi_{bv}}{\omega_{bv}}s + 1} \tag{13}$$

where $\omega_{bv}$ is the natural frequency, rad/s, with $\omega_{bv} = \sqrt{\frac{K_l + K_v + K_{fv}}{m_T + m_v}}$; $\xi_{bv}$ is the damping coefficient, with $\xi_{bv} = \frac{1}{2}\sqrt{\frac{(B_T + B_v)^2}{(m_T + m_v)(K_l + K_v + K_{fv})}}$; $K_{bv}$ is the gain coefficient, m/A, given by $K_{bv} = \frac{K_I}{(K_l + K_v + K_{fv})}$.

The hydraulic oil exiting the proportional directional valve is directed into the valve-controlled hydraulic cylinder. The mechanical schematic of this system is illustrated in Figure 4.

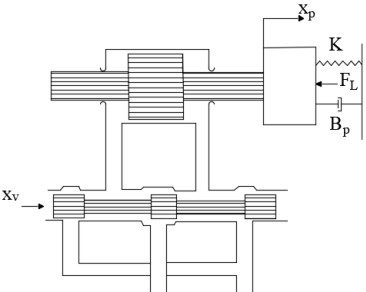

**Figure 4.** Schematic diagram of the valve-controlled hydraulic cylinder.

Assuming that the sliding valve is at zero opening and the four throttling ports are symmetrical, for convenience in calculation, the variations from the initial conditions are represented by the variables themselves. The Laplace transform of the linearized flow equation is given by Equation (14):

$$q_L(s) = K_q x_v(s) - K_c p_L(s) \tag{14}$$

where $q_L$ is the load flow rate, L/min, given by $q_L = \frac{q_1 + q_2}{2}$; $K_q$ is the flow amplification coefficient, with $K_q = \frac{\partial q_L}{\partial x_v}$; $x_v$ is the spool displacement, m; $K_c$ is the flow-pressure coefficient, given as $K_c = -\frac{\partial q_L}{\partial p_L}$.

The Laplace transform form of the continuity equation for the hydraulic cylinder's flow rate is shown in Equation (15) [18].

$$q_L(s) = C_{tp}p_L(s) + A_p s x_p(s) + \frac{V_t}{4\beta_e}sp_L(s) \tag{15}$$

where $V_t$ is the total volume of the two oil chambers of the flow adjustment cylinder, m$^3$; $C_{tp}$ is the total leakage coefficient of the flow adjustment cylinder; $A_P$ is the effective area of the flow adjustment cylinder, m$^2$; $\beta_e$ is the overall elastic modulus of the system, N/m$^3$.

The external load force acting on the flow adjustment cylinder generally consists of four components: inertia force, viscous damping force, elastic force, and other external loads [19]. The Laplace-transformed form of the resulting equilibrium equation is presented in Equation (16).

$$A_P p_L(s) = (m_p s^2 + B_p s + K)x_p(s) + F_L(s) \tag{16}$$

where $m_p$ is the total mass attributed to the piston component; $B_p$ is the viscous damping coefficient of the piston and load; $K$ is the spring stiffness of the load; $F_L$ is the external load force applied to the piston.

By combining Equations (14)–(16), the displacement of the flow adjustment cylinder piston, considering the combined effects of the proportional directional valve spool displacement and the external load force, is expressed in Equation (17).

$$x_p(s) = \frac{\frac{K_q}{A_P}x_v(s) - \frac{K_{ce}}{A^2_P}\left(1 + \frac{V_t}{4\beta_e K_{ce}}s\right)F_L(s)}{\frac{m_p V_t}{4\beta_e A^2_P}s^3 + \left(\frac{K_{ce}m_p}{A^2_P} + \frac{V_t B_p}{4\beta_e A^2_P}\right)s^2 + \left(\frac{K_{ce}B_p}{A^2_P} + \frac{V_t K}{4\beta_e A^2_P} + 1\right)s + \frac{K_{ce}K}{A^2_P}} \tag{17}$$

where $K_{ce}$ is the total flow pressure coefficient, where $K_{ce} = K_c + C_{tp}$.

In the preceding servo system, the presence of an elastic load is nearly negligible. Consequently, the parameters $K = 0$, $K_{ce}$, and $B_P$ can be regarded as very small, and their product is even smaller. As a result, the impact can be disregarded. The simplification of the preceding equation leads to the derivation of the transfer function between the valve spool displacement $x_v(s)$ and the displacement of the flow adjustment cylinder piston $x_p(s)$, as depicted in Equation (18), and the transfer function between the external load $F_L(s)$ and the displacement of the flow adjustment cylinder piston $x_p(s)$, as illustrated in Equation (19).

$$\frac{x_p(s)}{x_v(s)} = \frac{\frac{K_q}{A_p}}{s\left(\frac{s^2}{\omega_p^2} + \frac{2\delta_p}{\omega_p}s + 1\right)} \tag{18}$$

$$\frac{x_p(s)}{F_L(s)} = \frac{-\frac{K_{ce}}{A_p^2}\left(1 + \frac{V_t}{4\beta_e K_{ce}}s\right)}{s\left(\frac{s^2}{\omega_p^2} + \frac{2\delta_p}{\omega_p}s + 1\right)} \tag{19}$$

where $\omega_p = \sqrt{\frac{4\beta_e A_p^2}{V_t m_p}}$; $\delta_p = \frac{K_{ce}}{A_p}\sqrt{\frac{\beta_e m_p}{V_t}} + \frac{B_p}{4A_p}\sqrt{\frac{V_t}{\beta_e m_p}}$.

The push rod of the flow control lever is connected to the swashplate of the axial piston pump, enabling it to modify the swashplate angle through the extension and retraction of the push rod. The change in the swashplate angle is directly proportional to the piston rod displacement, as presented in Equation (20).

$$\psi = \frac{\psi_{max}}{X_{max}X_p} \tag{20}$$

where $\psi_{\max}$ is the maximum swashplate angle; $X_{\max}$ is the maximum displacement of the flow adjustment push rod.

Upon obtaining the transfer function of the hydraulic pump variable system, it is essential to derive the transfer function of the pump-controlled motor system. The following assumptions are required in this process:

(1) The input speed of the variable pump remains constant;
(2) The interconnecting pipeline between the variable pump and the fixed motor is short, disregarding pipeline losses. The hydraulic pipelines have identical characteristics, and the volume of the pump and motor chambers remains constant;
(3) The pressure within the variable pump and fixed motor chambers is atmospheric, and the leakage is laminar;
(4) All chamber pressures are equivalent, and the physical properties of the hydraulic fluid remain constant;
(5) The pressure of the replenishment system remains constant, and the low-pressure pipeline pressure during operation equals the system replenishment pressure;
(6) The input signal is minimal, and pressure saturation does not occur.

The displacement of the variable pump is also approximately proportional to the swashplate angle [19], which therefore gives the following:

$$D_B = K_B \psi \tag{21}$$

where $D_B$ is the variable pump displacement, m³/rad; $K_B$ is the displacement conversion factor.

The continuous flow equation of the variable pump is as follows:

$$q_B = D_B \omega_B - C_{iB}(p_h - p_l) - C_{eB} p_h \tag{22}$$

where $q_B$ is the flow output from the variable pump; $\omega_B$ is the variable pump speed; $C_{iB}$, $C_{eB}$ are the internal and external leakage coefficients of the variable pump; $p_h$ is the high-pressure pipeline pressure; $p_l$ is the low-pressure replenishment pipeline pressure.

After performing the Laplace transform on the above equation, we obtain the following:

$$Q_B(s) = K_{qB} \psi(s) - C_{tB} P_h(s) \tag{23}$$

where $K_{qp} = K_B \omega_B$; $C_{tB} = C_{iB} + C_{eB}$.

In the pump-controlled motor system, the flow output from the variable pump is primarily divided into three components: the flow entering the hydraulic motor, the motor's internal and external leakage, and the flow variation resulting from the compressibility of the oil. Consequently, Equation (24) presents the continuity equation for the high-pressure chamber of the hydraulic motor.

$$q_B = C_{im}(p_h - p_l) + C_{em} p_h + D_m \frac{d\theta_m}{dt} + \frac{V_0}{\beta_e} \frac{dp_h}{dt} \tag{24}$$

where $C_{im}$, $C_{em}$ are the internal and external leakage coefficients of the motor; $D_m$ is the motor displacement, m³/rad; $\theta_m$ is the motor swivel angle, rad; $\beta_e$ is the effective bulk modulus of elasticity, N/m³; $V_0$ is the total chamber volume, m³.

The Laplace transform of the above equation yields the following:

$$Q_B(s) = (C_{im} + C_{em}) P_h(s) + D_m s \theta_m(s) + \frac{V_0}{\beta_e} s P_h(s) \tag{25}$$

The output torque of the motor in the pump-controlled motor system is primarily utilized to counteract the motor's inertial force, the viscous resistance during rotation, and the external load torque applied to the motor shaft. Thus, Equation (26) represents the torque equilibrium equation for the hydraulic motor and the load.

$$D_m(p_h - p_l) = J_m \frac{d^2\theta_m}{dt^2} + B_m \frac{d\theta_m}{dt} + G_0\theta_m + T_L \tag{26}$$

where $J_m$ is the total rotational inertia of the hydraulic motor and the load, $kg/m^2$. $B_m$ is the viscous damping coefficient, $kg/s$. $G_0$ is the load torsional stiffness; $T_L$ is the external load torque on the fixed motor shaft.

The result after performing the Laplace transform is the following:

$$D_m P_h(s) = J_m s^2 \theta_m(s) + B_m s\theta_m(s) + G_0\theta_m(s) + T_L(s) \tag{27}$$

By combining Equations (23), (25) and (27), we derive the transfer function of the pump-controlled motor system, as illustrated in Equation (28).

$$\theta_m(s) = \frac{\frac{K_{qB}}{D_m}\psi(s) - \frac{C_t}{D_m^2}\left(1 + \frac{V_0}{C_t\beta_e}s\right)T_L(s)}{\frac{J_m V_0}{D_m^2\beta_e}s^3 + \left(\frac{J_m C_t}{D_m^2} + \frac{B_m V_0}{D_m^2\beta_e}\right)s^2 + \left(1 + \frac{B_m C_t}{D_m^2} + \frac{G_0 V_0}{D_m^2\beta_e}\right)s + \frac{C_t G_0}{D_m^2}} \tag{28}$$

where $C_t = C_{tB} + C_{im} + C_{em}$, which is the total leakage coefficient.

In this pump-controlled motor system, the variable mechanism is a servo system, and its load primarily consists of inertial loads [20]. Here, torsional stiffness can be disregarded, i.e., $G_0 = 0$. Generally, $D_m/C_t \gg B_m$; thus, the above equation can be simplified to the following:

$$\theta_m(s) = \frac{\frac{K_{qB}}{D_m}\psi(s) - \frac{C_t}{D_m^2}\left(1 + \frac{V_0}{\beta_e C_t}s\right)T_L(s)}{s\left(\frac{s^2}{\omega_h^2} + \frac{2\delta_h}{\omega_h}s + 1\right)} \tag{29}$$

where $\omega_h = \sqrt{\frac{\beta_e D_m^2}{V_0 J_m}}$; $\delta_h = \frac{C_t}{2D_m}\sqrt{\frac{\beta_e J_m}{V_0}} + \frac{B_m}{2D_m}\sqrt{\frac{V_0}{\beta_e J_m}}$.

By substituting $\dot{\theta}_m(s) = s\theta_m(s)$ into Equation (29), we can obtain the transfer function that relates the motor speed to the input variable, which is the swashplate angle of the variable pump.

$$\frac{\dot{\theta}_m(s)}{\psi(s)} = \frac{\frac{K_{qB}}{D_m}}{\frac{s^2}{\omega_h^2} + \frac{2\delta_h}{\omega_h}s + 1} \tag{30}$$

The transfer function of motor speed to the input value is obtained when the input variable is the external load on the motor shaft.

$$\frac{\dot{\theta}_m(s)}{T_L(s)} = \frac{-\frac{C_t}{D_m^2}\left(1 + \frac{V_0}{\beta_e C_t}s\right)}{\frac{s^2}{\omega_h^2} + \frac{2\delta_h}{\omega_h}s + 1} \tag{31}$$

To establish closed-loop control of the system, the motor speed must be continuously monitored and converted into a corresponding voltage value using a sensor. This voltage value is then compared with the desired value for the expected speed, obtaining an error signal that is subsequently transmitted to the controller through a proportional amplifier. Through adjustment of the error signal, the motor speed is consistently brought closer to the desired speed. The speed sensor functions as a proportional component, characterized by the following transfer function:

$$K_v = \frac{U(s)}{\dot{\theta}_m(s)} \tag{32}$$

In summary, Figure 5 depicts the transfer function block diagram of the pump-controlled motor speed control system for the monorail drive system.

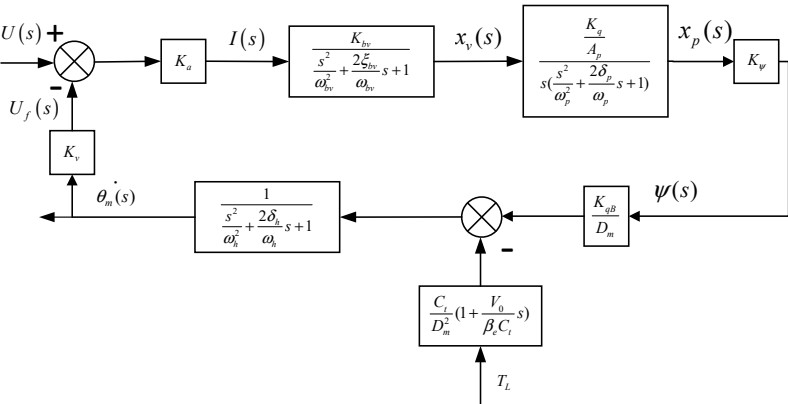

**Figure 5.** Transfer function block diagram of the pump-controlled motor system.

For the pump-controlled motor system, the dynamic response of the proportional variable pump and the fixed motor link is the least pronounced within the system, with a resonant frequency significantly lower than that of the proportional valve and servo cylinder. As a result, the dynamics of the latter two can be neglected [21]. The transfer function of the electro-hydraulic proportional directional valve can be denoted as shown in Figure 5. Additionally, the simplified transfer function of the valve-controlled cylinder can be condensed into an integral link and a proportional link [22], which leads to the illustration of the simplified system transfer function block diagram in Figure 6.

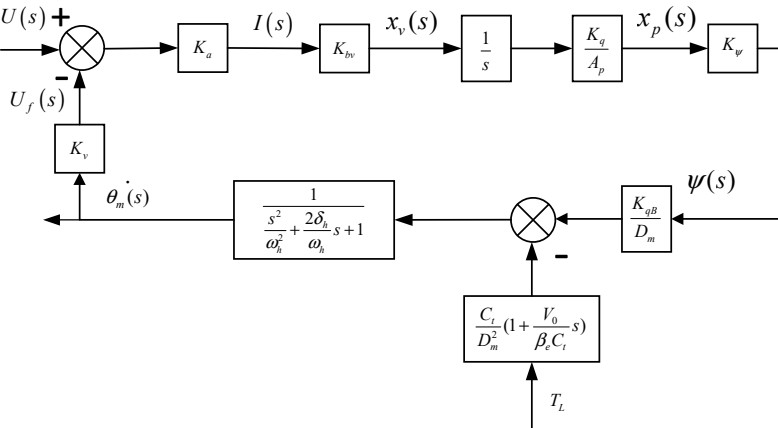

**Figure 6.** Simplified block diagram of the transfer function for the pump-controlled motor system.

Based on the aforementioned inference, the open-loop transfer function of the system is represented by the following equation:

$$G_k(s) = \frac{K_a K_{bv} K_q K_\psi K_{qB} K_v}{A_p D_m \left( \frac{s^3}{\omega_h^2} + \frac{2\delta_h}{\omega_h} s^2 + s \right)} \tag{33}$$

The closed-loop transfer function of the system, when the input signal is voltage $U(s)$, is expressed as follows:

$$G_B(s) = \frac{\dot{\theta_m}(s)}{U(s)} = \frac{K_g}{s \left( \frac{s^2}{\omega_h^2} + \frac{2\delta_h}{\omega_h} s + 1 \right) + K_{vx}} \tag{34}$$

where $K_{vx} = \frac{K_a K_{bv} K_q K_\psi K_{qB} K_v}{A_p D_m}$, $K_g = \frac{K_a K_{bv} K_q K_\psi K_{qB}}{A_p D_m}$, and $K_{vx} = K_g K_v$.

The values of the parameters related to the transfer function, as obtained from the Rexroth series variable displacement pump manual and parameter specifications, are presented in Table 2.

**Table 2.** Parameters related to the monorail crane speed control transfer function.

| Parameters | Values | Parameters | Values |
|:---:|:---:|:---:|:---:|
| $K_a$ | 0.2 (A/V) | $V_0$ | $3.6 \times 10^{-2}$ (m$^3$) |
| $K_{bv}$ | 0.251 | $C_t$ | $5 \times 10^{-12}$ (m$^5$/N·s) |
| $K_q$ | 0.1783 (m$^2$/s) | $J_m$ | 0.052 (kg/m$^2$) |
| $A_p$ | $3.14 \times 10^{-2}$ (m$^2$) | $D_m$ | $9.0 \times 10^{-5}$ (m$^3$/rad) |
| $K_\psi$ | 5 | $\beta_e$ | $6.98 \times 10^8$ (N/m$^3$) |
| $K_{qB}$ | $1.756 \times 10^{-2}$ (m$^3$/rad·s) | $B_m$ | 3.26 (N·m·s) |
| $K_g$ | 280 | $\omega_h$ | 55 (rad/s) |
| $K_v$ | 0.15 | $\delta_h$ | 0.57 |
| $K_{vx}$ | 42 | | |

The above parameters are used to obtain the open-loop transfer function of the system.

$$G_k(s) = \frac{42}{3.3 \times 10^{-4}s^3 + 2.1 \times 10^{-2}s^2 + s} = \frac{125,980}{s^3 + 62.79s^2 + 3020s} \tag{35}$$

The closed-loop transfer function is the following:

$$G_B(s) = \frac{280}{3.3 \times 10^{-4}s^3 + 2.1 \times 10^{-2}s^2 + s + 42} = \frac{848485}{s^3 + 62.79s^2 + 3020s + s + 127273} \tag{36}$$

The Bode plot of the open-loop transfer function, shown in Figure 7, indicates that the system's gain crossover frequency is below the phase crossover frequency, with both the phase margin and gain margin being positive. Thus, the system is stable and controllable.

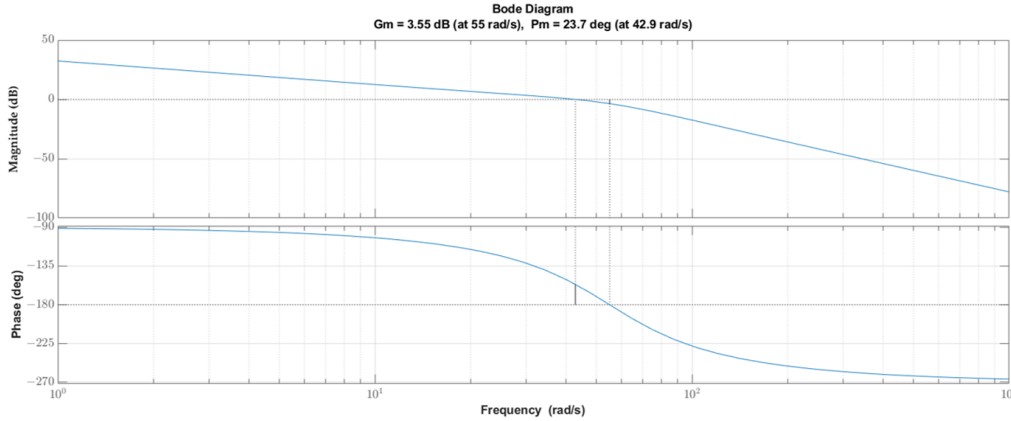

**Figure 7.** Open-loop Bode plot of the system.

## 4. Research on Optimization of Speed Control Characteristics of Monorail Crane Based on AMESim

This study analyzed the mathematical models of various loops in the speed regulation circuit of the monorail crane hydraulic system and derived the corresponding transfer functions. To create a more intuitive simulation model of the monorail crane power system, AMESim (v.21.1) simulation software can be used to develop this model based on the established mathematical model.

The experimental hydraulic system configuration in this paper utilizes an A4VG280 variable pump which is produced by Bosch Rexroth AG located on Glockeraustraße 4 89275 Elchingen, Germany. The main pump has a maximum displacement of 280 mL/r, while the slippage pump has a maximum displacement of 60 mL/r. The system is powered by an explosion-proof diesel engine rated, at 160 kW horsepower, running at 2100 RPM, and a fixed displacement motor with a displacement of 565 mL/r. Subsequently, an AMESim simulation model of the monorail crane drive speed regulation circuit is developed based on these parameters.

### 4.1. Modeling of Variable Displacement Pump Supply Module

As a crucial element of the speed regulation system for pump-controlled motors, it is essential to precisely simulate the variable displacement pump's control over the movement of the variable oil cylinder. This control is achieved through a proportional valve, ultimately impacting the swash plate angle and fine-tuning the pump's output displacement. In this study, the simulation model of the variable displacement pump supply module is developed leveraging the integrated capabilities of the HCD, mechanical, and signal libraries in the AMESim software, as depicted in Figure 8.

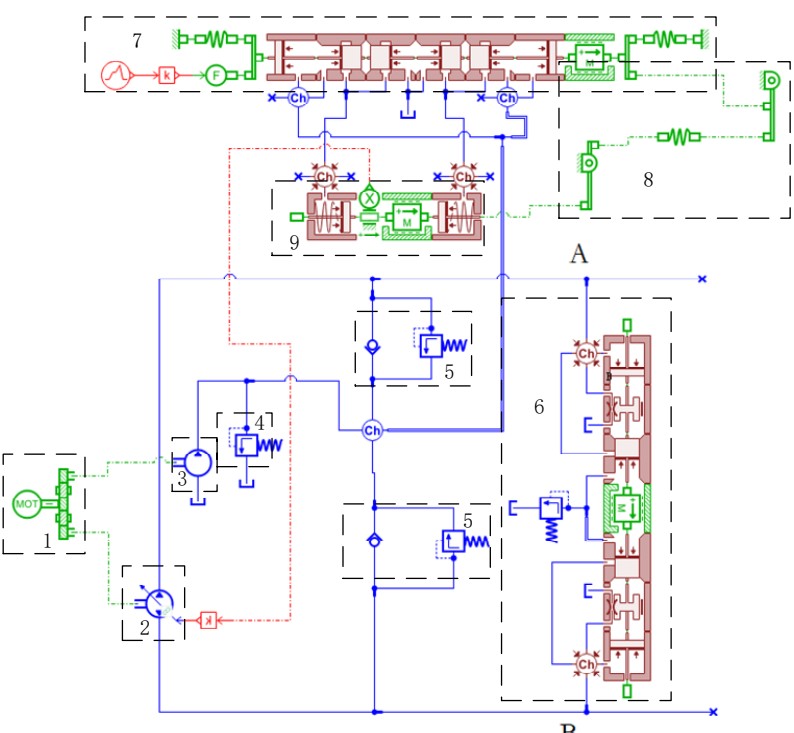

**Figure 8.** AMESim simulation model of the variable displacement pump supply module. 1—Diesel engine; 2—radial piston pump; 3—slippage pump; 4—low-pressure relief valve; 5—high-pressure relief valve group; 6—flush valve group; 7—proportional valve; 8—fork mechanism; 9—flow control cylinder; A—Interface of Oil Route A; B—Interface of Oil Route B.

The relationship between the proportional valve input current signal and the variable pump output displacement can be found in Figure 9 of the A4VG pump manual.

When the current at terminal A of the solenoid valve ranges from 200 to 400, the variable pump displacement changes from 0 to the maximum. Conversely, as the current at terminal A ranges from −400 to −200, the variable pump reverses the output displacement from the maximum to 0.

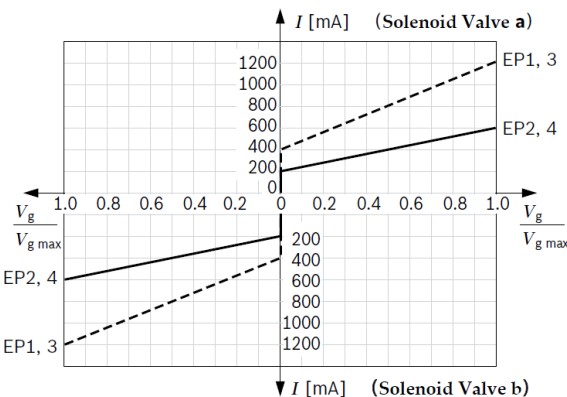

**Figure 9.** Relationship between the proportional valve current signal and pump displacement.

### 4.2. Modeling of Load Module

The monorail crane's operational speed varies from 0.5 to 2 m/s under different conditions. The simulation model depicts the load end with two drive units, comprising four drive motors, as illustrated in Figure 10.

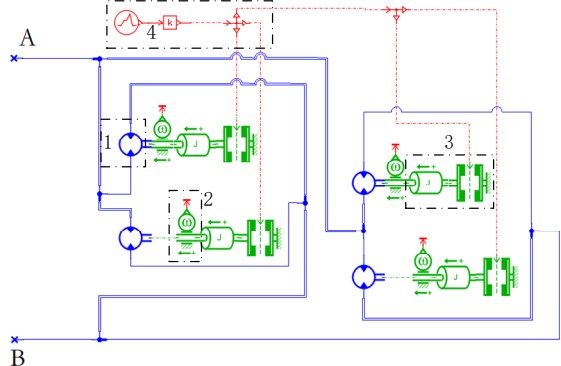

**Figure 10.** Relationship between the proportional valve current signal and pump displacement. 1—Fixed displacement motor; 2—speed sensor; 3—load application module; 4—load signal input; A—Interface of Oil Route A; B—Interface of Oil Route B.

In the simulation model, the fixed displacement motor is connected to the torque and friction modules in the AMESim mechanical library, simulating the process in which the drive friction wheel, installed on the drive motor of the monorail crane, provides traction by rolling contact with the rail under the action of clamping force. The load signal input terminal inputs the clamping force value matched to the operating conditions, with its simulation calculation module as depicted in Figure 11. The speed sensor provides real-time motor speed feedback to the controller.

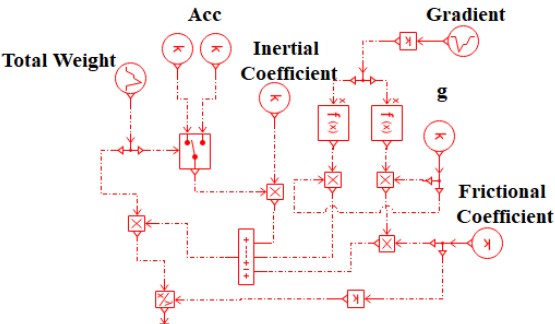

**Figure 11.** Clamping force calculation module.

### 4.3. Main Speed Control Loop Simulation Model

The primary speed control loop of the monorail crane consists of a variable pump and several fixed displacement motors. The simulation model for the entire loop was built using AMESim simulation software, as depicted in Figure 12.

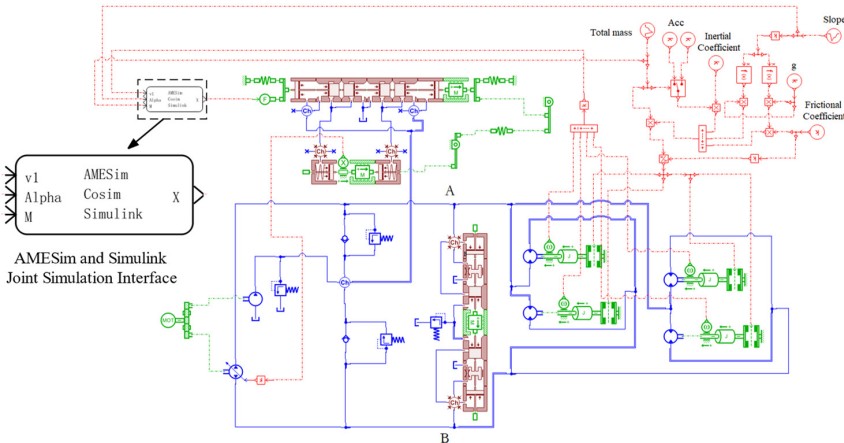

**Figure 12.** Main speed control loop simulation model. A—Oil Route A; B—Oil Route B.

In the above figure, the dashed box represents the interface for the joint simulation of AMESim and Simulink. It includes three input interfaces: the v1 interface input represents the average real-time output value of the speed sensors for the four drive motors, the Alpha interface input represents the pitch angle of the monorail crane operation, and the M interface input represents the total weight of the monorail crane. The output comprises the value of the proportional valve input electrical signal, enabling the joint simulation to achieve automatic speed control of the simulation model.

### 4.4. Speed Control Strategy

The main speed control loop of the monorail crane hydraulic system is a complex nonlinear system, influenced by the rheological properties of the hydraulic fluid, load variability, and the precision of the model. These factors impact the dynamic, static performance, and control accuracy of the pump-controlled motor speed regulation loop. The common control methods utilized in pump-controlled motor simulation research include traditional PID control, fuzzy control, and fuzzy adaptive PID control. Assuming the monorail crane operates under clamping force requirements, varied control algorithms are integrated into Simulink for speed regulation of the AMESim simulation model through the joint simulation interface.

We assume an input target of one for the open-loop transfer function, and its response curve is shown in Figure 13. The overshoot is 50%, and the settling time is 0.7 s.

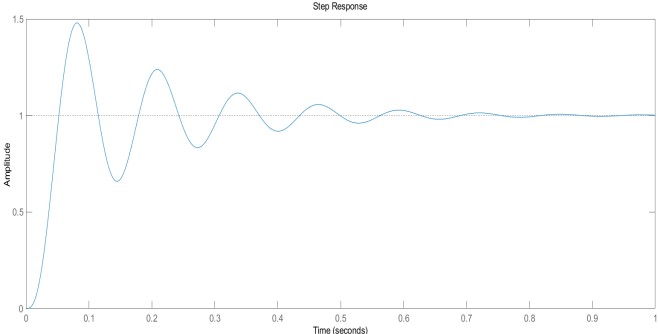

**Figure 13.** Open-loop transfer function step response plot (input is 1).

### 4.4.1. Traditional PID Control

The PID controller utilizes feedback to measure the error signal and regulate the controlled object based on this signal. It integrates three components: proportional, integral, and derivative. The control equation is described as follows:

$$u(t) = K_p e(t) + K_i \int_0^t e(\tau) d\tau + K_d \frac{de(t)}{dt} \tag{37}$$

In the simulation software, the compilation of the PID controller is illustrated in Figure 14.

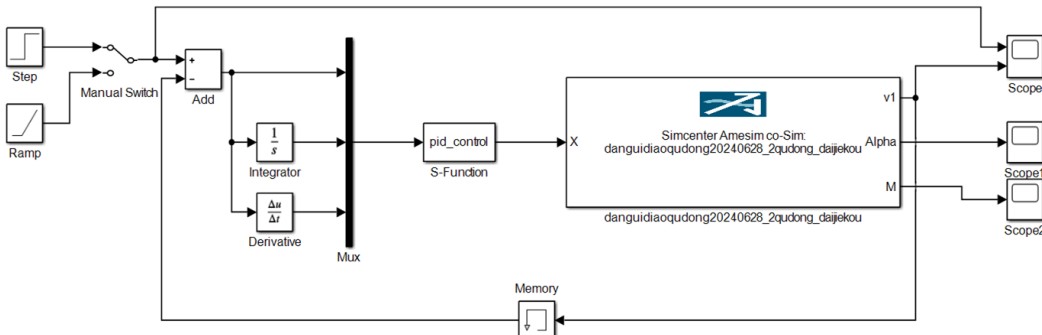

**Figure 14.** Design diagram of PID speed controller.

As an illustration, taking the flat and empty transport condition of the monorail crane as an example with a target vehicle speed of 2.0 m/s, the motor speed is approximately 112 rpm, calculated based on the drive wheel diameter of 340 mm. The control initiates 1 s after the monorail crane is ready to depart. At this point, the gradient signal in AMESim is set to 0 and the total weight signal is set to 15,000.

Critical Proportional Method PID Tuning

The PID parameters are tuned using the critical proportionality method. A step signal is applied to the open-loop transfer function of the system, as defined by Equation (35). By adjusting only the proportional gain, the system's response achieves a state of sustained oscillation, yielding a critical proportional gain $K_{ps} = 1.506$ and an oscillation period $T_r = 0.114$ s. The initial values of the three parameters of the classical PID controller are calculated according to the rules outlined in Table 3.

**Table 3.** Critical proportionality method PID parameters table.

| Type | $K_p$ | $K_i$ | $K_d$ |
|:---:|:---:|:---:|:---:|
| P | 0.5 $K_{ps}$ | | |
| PI | 0.45 $K_{ps}$ | 0.54 $K_{ps}/T_r$ | |
| PID | 0.6 $K_{ps}$ | 1.2 $K_{ps}/T_r$ | 0.075 $K_{ps}T_r$ |

Simulation results demonstrate that the control effectiveness improves when utilizing the PID controller. The step response is shown in Figure 15, and the PID parameters are derived from the calculations presented in the Table 3.

The figure indicates that, under the influence of a step signal of 112 rpm, the maximum overshoot is 28.4%, and the time required to reach steady state is 0.52 s.

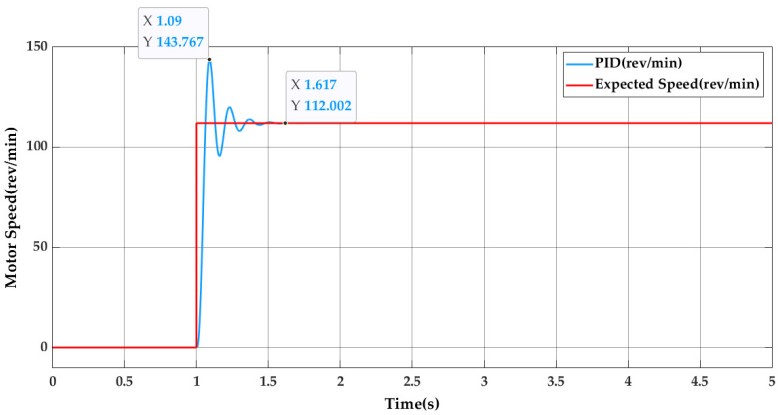

**Figure 15.** Speed control effect diagram of the PID control algorithm.

Root Locus Method for PID Tuning

The root locus method tunes the PID controller by specifying performance indices related to the system's frequency response. The desired index is defined as overshoot $\sigma \leq 10\%$ and settling time criteria $t_s = 0.3$ s.

The Bode plot analysis of the open-loop transfer function (Figure 7) yields a gain margin of $G_m = 3.55$ dB and a phase margin of $P_m = 23.7385°$, corresponding to frequencies $\omega_{cG} = 54.9545$ and $\omega_{cP} = 42.8558$, respectively. The root locus of the closed-loop transfer function is depicted in Figure 16.

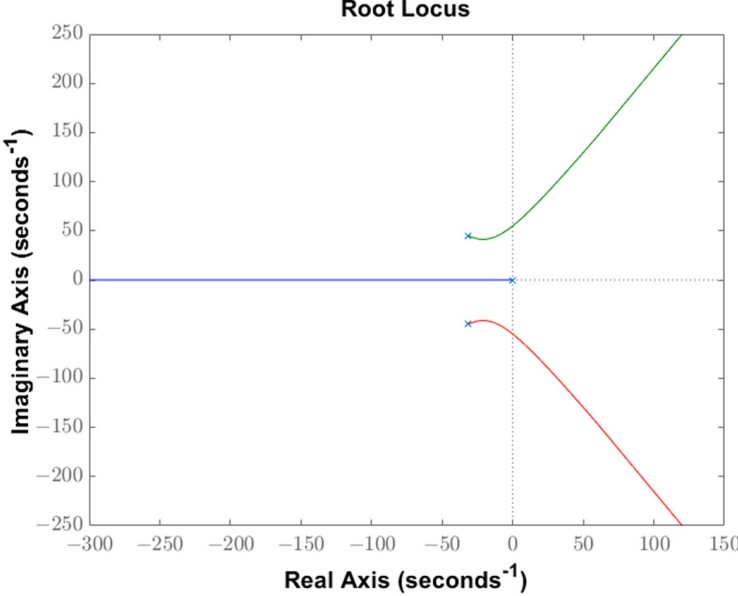

**Figure 16.** Root locus diagram of closed-loop function.

Analysis of the open-loop transfer function reveals that the system is a Type 1 system with a modest phase margin. Recognizing that the addition of an integral term would raise the system's order and potentially impact its stability, we opt to omit the integral term initially and concentrate on PD control.

When a PD controller with parameters $k(\tau s + 1)$ is introduced, with $k = 1$ and $\tau = 42$, the resulting root locus and step response is depicted in Figure 17.

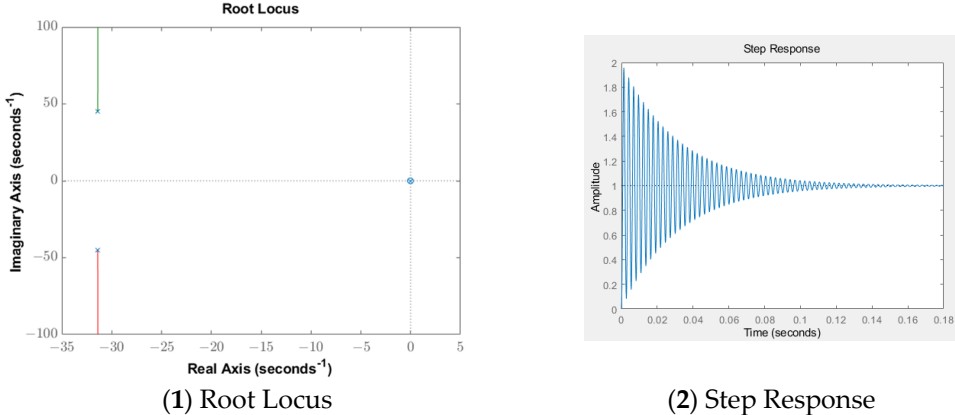

(**1**) Root Locus            (**2**) Step Response

**Figure 17.** Root locus plot and step response plot with added PD controller ($k = 1$, $\tau = 42$).

By substituting ($k = 1$, $\tau = 1/30$) and ($k = 0.01$, $\tau = 1/30$), the step response plots are displayed in Figure 18(1,2), correspondingly. When $k > 1$, the system response is fast, but overshoot cannot be reduced. When $k < 1$, although overshoot is suppressed, the response time becomes slower (as show in Figure 17), especially when k is minimized, leading to potential steady-state errors.

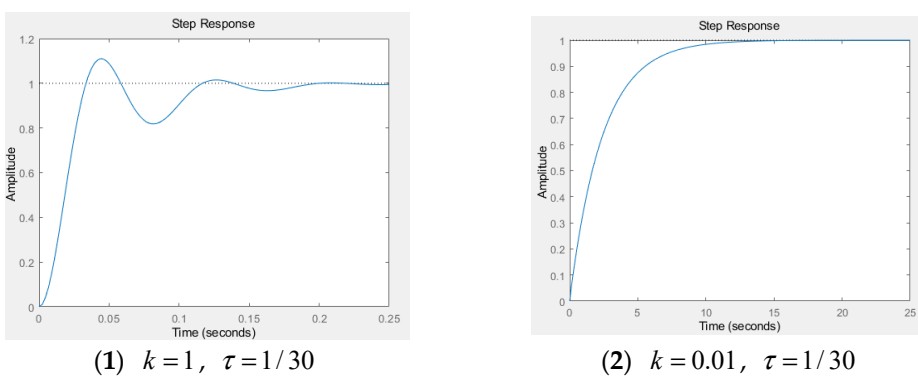

(**1**)   $k = 1$,   $\tau = 1/30$        (**2**)   $k = 0.01$,   $\tau = 1/30$

**Figure 18.** Step response plot of system with added PD controller.

Building on the previous results, we propose adding an integral term to the controller. The controller structure is defined as follows: $k/s \, (\tau_1 s + 1)(\tau_2 s + 1)$, with ($k = 1$, $\tau_1 = 1/30$, $\tau_2 = 1/0.1$). The modified root locus and step response are presented in Figure 19. At this stage, the overshoot is too large, prompting the testing of a proportional gain ($k < 1$). Ultimately, the values ($k = 1$, $\tau_1 = 1/30$, $\tau_2 = 1/0.1$) are selected. The step response is shown in Figure 20, where the overshoot is 3% and the settling time is 0.232 s.

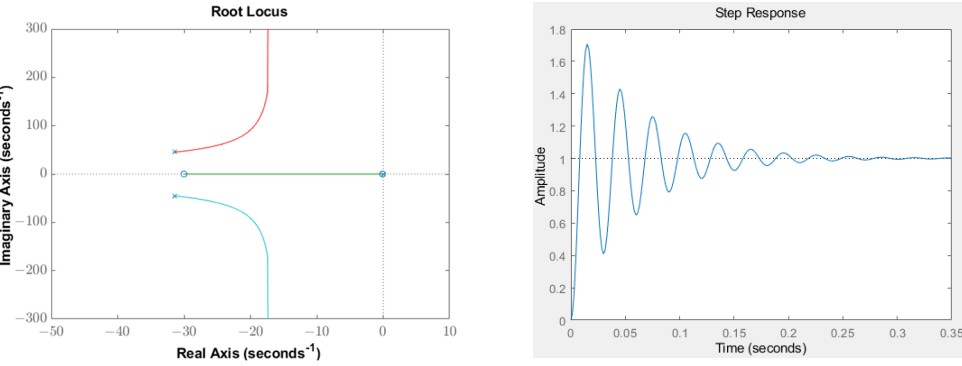

**Figure 19.** Root locus plot and step response plot with added PD controller ($k = 1$, $\tau_1 = 1/30$, $\tau_2 = 1/0.1$).

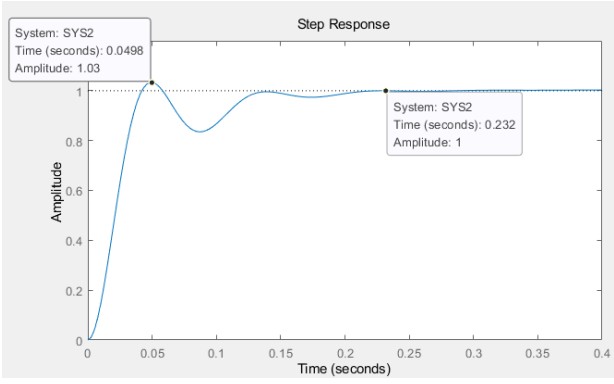

**Figure 20.** Step response plot with added PD controller ($k = 0.08, \tau_1 = 1/30, \tau_2 = 1/0.1$).

At this stage, the controller's transfer function corresponds to Equation (38).

$$G_c(s) = \frac{0.08}{s}\left(\frac{1}{30}s + 1\right)\left(\frac{1}{0.1}s + 1\right) \tag{38}$$

By expanding the formula, the corresponding PID coefficients are determined as $K_p = 0.803$, $K_i = 0.08$, and $K_d = 0.027$.

A Modification of the PID Law

In commercial control systems, the traditional PID control law as Equation (37) can be adjusted to use the process variable as input for derivative and proportional control, while the error solely serves as the input for integral control. This adjustment transforms the control law into Equation (39).

$$u(s) = -Kp * \dot{\theta}_m(s) - Kd * s * \dot{\theta}_m(s) + \left(\frac{Ki}{s}\right) * e(s) \tag{39}$$

Here, variable $\dot{\theta}_m(s)$ acts as the process variable, input into the proportional and derivative stages, while error $e(s)$ continues to be input into the integral component. Using the control law discussed above, the simulation model diagram developed in Simulink is shown in Figure 21.

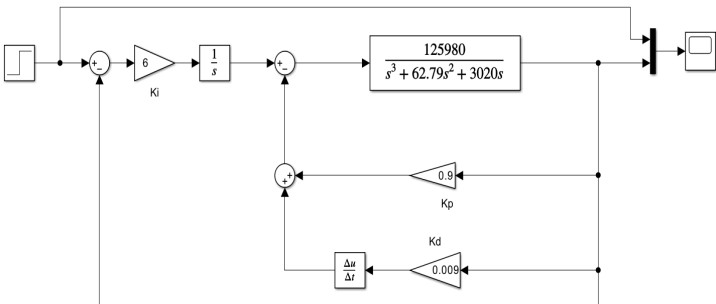

**Figure 21.** Simulink simulation block diagram.

We systematically tuned the three PID parameters by first adjusting $K_p$, followed by $K_i$ and $K_d$ in sequence, and observed the system's response to a step input with a target value of one. During the adjustment of $K_p$, we observed that when $K_p > 1$, the system response became unstable and diverged as $K_p$ increased. In contrast, when $K_p < 1$, further reduction in $K_p$ led to an increase in settling time. Therefore, the optimal value of $K_p$ was determined to be slightly less than 1. With $K_p$ fixed, increasing $K_i$ shortened the settling time; however, when $K_i$ exceeded six, overshoot occurred, and the improvement in settling time diminished, with the response curve beginning to exhibit oscillations. As a result, $K_i$

was set to six. Finally, by adjusting $K_d$, the response curve was further smoothed. The three parameters were ultimately set to ($K_p = 0.9$), ($K_i = 6$), and ($K_d = 0.09$), and the system's response curve under this controller is shown in Figure 22.

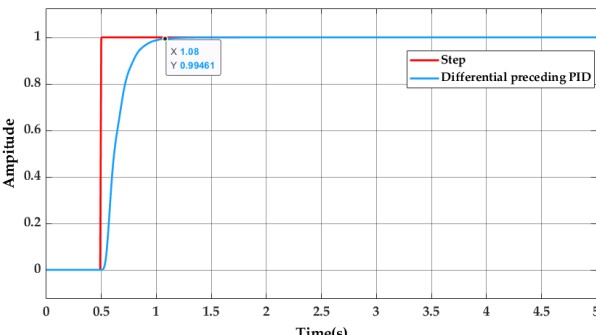

**Figure 22.** Step response plot.

As shown in the graph, by incorporating the aforementioned controller, the system exhibits negligible overshoot in response to a step input with an amplitude of one and a settling time of 0.58 s, demonstrating excellent overshoot suppression.

Conclusion of the Design and Tuning of the Traditional PID Controller

Comparing various PID controller design approaches and tuning strategies discussed earlier, we infer that the classical PID controller crafted through the critical ratio method shows slightly inferior performance compared to those tailored using the root locus method and incorporating process parameter inputs. The PID controller incorporating process parameter inputs excels in overshoot suppression, while the one fine-tuned via the root locus method demonstrates swifter settling time response. The root locus method, capable of limiting overshoot to approximately 3% and ensuring rapid step response, is selected as the definitive tuning technique for the conventional PID controller in forthcoming comparative analyses.

### 4.4.2. Fuzzy Adaptive PID Control

The traditional PID controller previously mentioned performs well in tracking the target speed of the monorail crane speed control loop. Nevertheless, its PID controller parameters cannot autonomously adapt to the intricate and fluctuating operational conditions and the adaptive control requisites of the monorail crane. The fuzzy adaptive PID algorithm integrates a fuzzy controller and a PID controller. The fuzzy controller uses the error and its rate of change as inputs and dynamically adjusts the PID controller parameters using fuzzy rules, ensuring that the controlled object maintains good dynamic and static stability.

For fuzzy adaptive PID control, fuzzy rules must first be formulated. Utilizing these fuzzy rules, the error and the rate of change of the error are subjected to fuzzification to obtain corresponding membership degrees. Then, in conjunction with the appropriate control rules, $\Delta K_p$, $\Delta K_i$, and $\Delta K_d$ are calculated, and are subsequently integrated into the PID controller for operation. The control expression is depicted in Equation (40).

$$u = (K_p + \Delta K_p)e(k) + (K_i + \Delta K_i)\sum_{n=0}^{k} e(n) + (K_d + \Delta K_d)(e(k) - e(k-1)) \qquad (40)$$

(1)    Determining input and output fuzzy variables and membership functions

The inputs of this controller are the motor speed deviation $e$ and the rate of change of the deviation $e_c$. The basic domain for the speed deviation is set to $[-50, 50]$, while the basic domain for the rate of change is set to $[-10, 10]$. Correspondingly, the fuzzy domain is defined as $E \in [-6, 6]$, with a quantization factor $G_e = 6/50 = 0.12$. The fuzzy domain for the rate of change is defined as $E_c \in [-3, 3]$, with a quantization factor $G_{ec} = 3/10 = 0.3$.

Based on the results of PID parameter tuning using the critical proportionality method, the fuzzy domain for the adjustment coefficients of the PID parameters is set to $\Delta K_p = [-3\ 3]$, $\Delta K_i = [-20\ 20]$, and $\Delta K_d = [-0.0005\ 0.0005]$. The fuzzy subsets for inputs $e$ and $e_c$ and outputs $\Delta K_p$, $\Delta K_i$, and $\Delta K_d$ are defined as $\{NB, NM, NS, ZO, PS, PM, PB\}$, all utilizing triangular membership functions.

The above parameters, combined with the experience of experts and technicians, lead to the derivation of 49 fuzzy rules, as shown in Figure 23 below:

(2)    Simulink simulation module construction

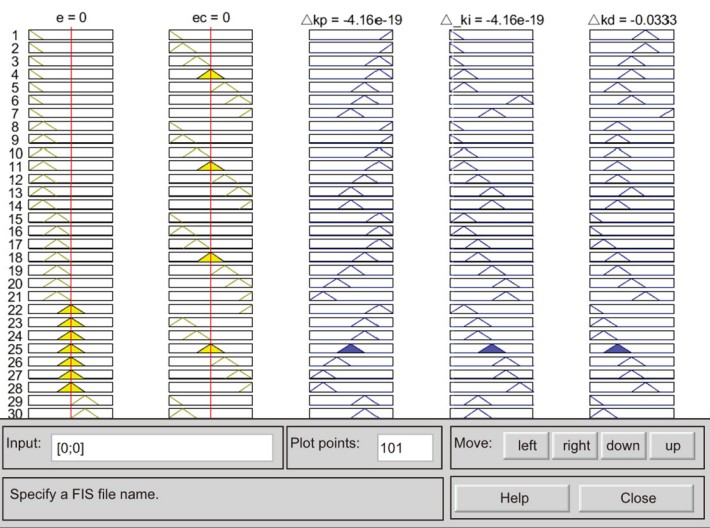

**Figure 23.** Fuzzy rules diagram.

After establishing the fuzzy rules, a PID controller structure with fuzzy adaptive components was constructed in Simulink using the co-simulation interface, as shown in Figure 24:

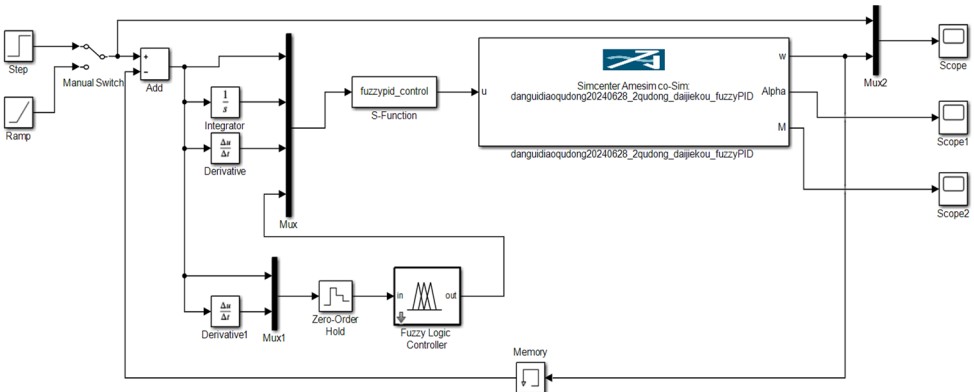

**Figure 24.** Design diagram of the fuzzy adaptive PID controller.

### 4.4.3. MPC Controller

Model Predictive Control (MPC) is an advanced control strategy that integrates process modeling, online optimization, and a rolling time horizon to compute a sequence of control inputs [23]. The essence of MPC is its use of a process model to predict future system behavior, allowing for the determination of a series of control actions by optimizing a cost function designed to guide the system along a desired trajectory.

The formula of the MPC controller is in the form of Equation (41):

$$x_{k+1} = Ax_k + Bu_k \tag{41}$$

Here, $(x_{k+1})$ represents the system state at time $(k + 1)$, $(x_k)$ denotes the system state at time $(k)$, and $(u_k)$ refers to the system input at time $(k)$. The forms of the system state matrix $(x_k)$ and the input matrix $(u_k)$ are provided in Equation (42).

$$X_k = \begin{bmatrix} x(k|k) \\ x(k+1|k) \\ x(k+2|k) \\ \ldots \\ x(k+N|k) \end{bmatrix} \quad U_k = \begin{bmatrix} u(k|k) \\ u(k+1|k) \\ u(k+2|k) \\ \ldots \\ u(k+N-1|k) \end{bmatrix} \tag{42}$$

Here, $(x(k + i|k))$ represents the predicted state at time $(k + i)$, based on the state at time $(k)$, and $(u(k + i|k))$ represents the predicted input at time $(k + i)$, based on the input at time $(k)$. We then assume that the system input at this time is $(y = x)$, with the reference $(R = 0)$, and the error $(E = y - R = x)$.

Define the cost function (objective function in optimization) $J$ as follows:

$$J = \sum_{i=0}^{N-1} \left( E(k+i|k)^T Q E(k+i|k) + u(k+i|k)^T R u(k+i|k) \right) + E(k+N)^T F E(k+N) \tag{43}$$

where $(E(k+i|k)^T Q E(k+i|k))$ is the weighted sum of the errors, and $(u(k+i|k)^T R u(k+i|k))$ is the weighted sum of the inputs. Considering that $(E = x)$, we substitute $(E)$ in the above expression to obtain the new cost function:

$$J = \sum_{i=0}^{N-1} \left( x(k+i|k)^T Q x(k+i|k) + u(k+i|k)^T R u(k+i|k) \right) + x(k+N)^T F x(k+N) \tag{44}$$

where Q and R are both diagonal matrices. Expanding the above equation with terms involving x, we obtain its matrix form:

$$\begin{bmatrix} x(k|k) \\ x(k+1|k) \\ \ldots \\ x(k+N|k) \end{bmatrix}^T \begin{bmatrix} Q & & & \\ & Q & & \\ & & \ldots & \\ & & & F \end{bmatrix} \begin{bmatrix} x(k|k) \\ x(k+1|k) \\ \ldots \\ x(k+N|k) \end{bmatrix} \tag{45}$$

And substituting $\begin{bmatrix} Q & & & \\ & Q & & \\ & & \ldots & \\ & & & F \end{bmatrix} = \overline{Q},$ $\begin{bmatrix} x(k|k) \\ x(k+1|k) \\ \ldots \\ x(k+N|k) \end{bmatrix} = X_k$ into the equation, the

final form of the expansion with respect to x is $X_k^T \overline{Q} X_k$ Similarly, the final form of the expansion with respect to u is $U_k^T \overline{R} U_k$, and the cost function is rewritten as follows:

$$J = X_k^T \overline{Q} X_k + U_k^T \overline{R} U_k \tag{46}$$

In Equation (41), let the initial condition be $x(k|k) = x_k$, and iteratively compute it, yielding the following:

$$x(k|k) = x_k$$
$$x(k+1|k) = Ax(k|k) + Bu(k|k) = Ax_k + Bu(k|k)$$
$$x(k+2|k) = Ax(k+1|k) + Bu(k+1|k) = A^2 x_k + ABu(k|k) + Bu(k+1|k)$$
$$\ldots$$
$$x(k+N|k) = A^N x_k + A^{N-1} Bu(k|k) + \ldots + Bu(k+N-1|k) \tag{47}$$

Convert the above equation into matrix form, and substitute the value of the following expression into it.

$$\begin{bmatrix} I \\ A \\ A^2 \\ \dots \\ A^N \end{bmatrix} = M, \quad \begin{bmatrix} 0 & 0 & \dots & 0 \\ B & 0 & \dots & 0 \\ AB & B & \dots & 0 \\ \dots & \dots & \dots & 0 \\ A^{N-1} & A^{N-2}B & \dots & B \end{bmatrix} = C \tag{48}$$

Equation (47) can be rewritten as $X_k = Mx_k + CU_k$. It can be observed that the state matrix $X$ is now expressed in terms of the initial condition and input. Substituting this into Equation (46) yields the final form of the cost function.

$$J = x_k^T G x_k + 2 x_k^T E U_k + U_k^T H U_k \tag{49}$$

It can be observed that the cost function has been transformed into a function dependent on the initial conditions and inputs, which satisfies the standard form of a quadratic programming problem. Consequently, the input that minimizes the objective function, i.e., the optimal input, can be determined.

The Simulink toolbox features an MPC controller module. In this simulation, the sampling time is set to 0.1 s, the prediction horizon is set to 10, the control horizon is set to 2, the input weight is configured to 0.09, and the output weight is configured to 22.5.

Upon configuration, the module is integrated into the simulation model, as shown in Figure 25.

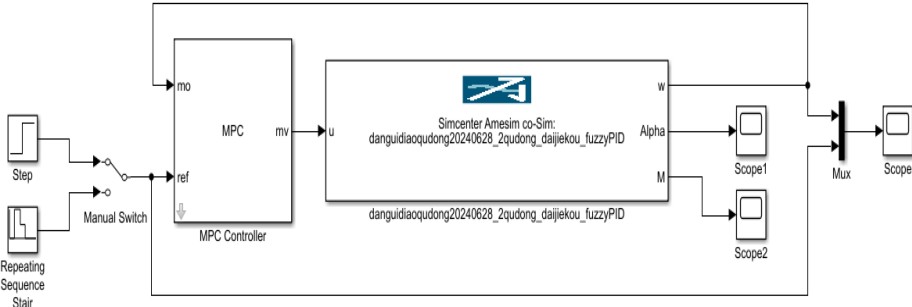

**Figure 25.** Design diagram of the MPC controller.

## 5. Analysis of Simulation Results

### 5.1. Step Velocity Response

As an example, traditional PID and fuzzy adaptive PID control methods are employed to regulate the hydraulic speed control system model of the monorail crane. Considering the empty-load transportation condition of a monorail crane with a target motor speed of 112 rpm, the simulation results are illustrated in Figure 26. Specific parameter comparisons can be found in Table 4.

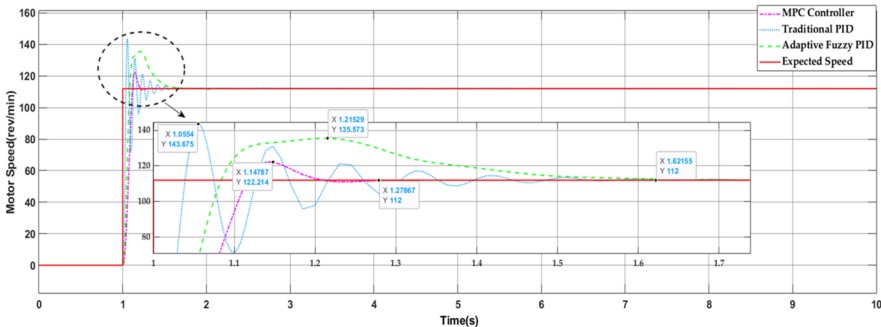

**Figure 26.** Comparison chart of step response effects for the 2 m/s speed control model in level tunnel.

**Table 4.** Comparison table of control effects.

| Method | Overshoot | Steady-State Time |
|---|---|---|
| PID | 27.6% | 0.62 s |
| Fuzzy PID | 20.5% | 0.62 s |
| MPC | 8.9% | 0.27 s |

Based on the statistical data, it is evident that, while all three controllers reach the steady state at the same time and their overshoot indicators are less than 30% of the control target, a notable variation exists in these indicators. The MPC controller demonstrates the smallest overshoot and the smoothest control curve, leading to the most optimal control effect.

### 5.2. Response to Changes in Target Speed

During its operation, the monorail crane encounters diverse operational conditions, necessitating dynamic adjustments to its travel speed. This study utilizes a series of continuous speed change processes as the control objective to evaluate the stability of speed regulation across different controllers under varied operational conditions. Using the signal generator in Simulink, the entire operation of the monorail crane is simulated, including acceleration to 2 m/s after 10 s, 10 s of travel, deceleration to 0.9 m/s, maintaining that speed for 10 s, and eventual deceleration to a stop, as illustrated in the simulation results presented in Figure 27.

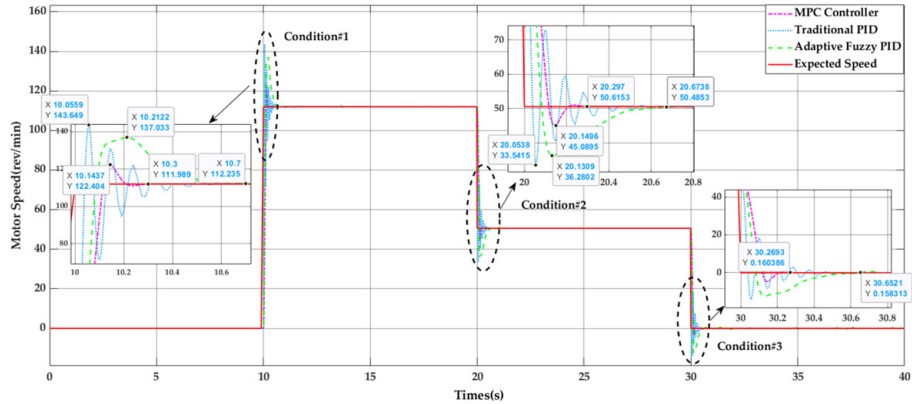

**Figure 27.** Comparison chart of response effects for the multi-condition speed control model.

Specific parameter comparisons can be found in Tables 5–7.

**Table 5.** Comparison table of control effects (Condition #1).

| Method | Overshoot | Steady-State Time |
|---|---|---|
| PID | 27.6% | 0.7 s |
| Fuzzy PID | 22.3% | 0.7 s |
| MPC | 8.9% | 0.3 s |

**Table 6.** Comparison table of control effects (Condition #2).

| Method | Overshoot | Steady-State Time |
|---|---|---|
| PID | 33.8% | 0.66 s |
| Fuzzy PID | 28.8% | 0.66 s |
| MPC | 11% | 0.3 s |

**Table 7.** Comparison table of control effects (Condition #3).

| Method | Steady-State Time |
| --- | --- |
| PID | 0.65 s |
| Fuzzy PID | 0.65 s |
| MPC | 0.26 s |

The statistical data reveal that during the continuous speed change processes within Conditions #1 and #2, the traditional PID controller no longer meets the control performance requirements in terms of overshoot indicators. While both the fuzzy PID and MPC controllers perform similarly in steady-state parameters, the MPC controller exhibits superior overshoot indicators compared to the fuzzy PID in both acceleration and deceleration conditions. Furthermore, during the deceleration to a stop in Condition #3, the MPC process demonstrates smoother control with minimal oscillation, ultimately achieving the most effective control.

## 6. Conclusions

A fast and precise speed control method is crucial for the efficient operation of a monorail crane. Consequently, this paper proposes an optimization approach for the speed regulation characteristics of the hydraulic speed control loop of monorail cranes. Initially, distinct operational conditions of monorail cranes are identified, and an analysis of the driving mechanism structure and load characteristics is conducted to establish the relationship between the traction force of the monorail crane and the hydraulic system parameters. Subsequently, a mathematical model of the hydraulic speed control loop of the monorail crane is presented, deriving the transfer function from the electromagnetic signal of the proportional valve to the variable pump swashplate angle change and to the hydraulic motor output speed, while ensuring its stability. Employing the mathematical model and hydraulic schematic, the monorail crane speed control loop is modeled using AMEsim software and jointly simulated with Simulink to probe the dynamic characteristics of the speed control loop under the influence of traditional PID controllers, fuzzy adaptive PID controllers, and MPC controllers. The efficacy of the self-designed fuzzy adaptive PID controller and MPC controller in the speed control process in this model is verified, with the MPC controller demonstrating an overshoot of 8.9% in the step response, a maximum overshoot of 11% during target speed changes, and a steady-state time of 0.26–0.3 s. Its dynamic characteristics in speed control surpass those of the other two methods. The collective simulation results furnish a theoretical foundation for achieving adaptive control and power matching of the monorail crane speed control loop. Furthermore, owing to the broad applicability of the pump-controlled motor speed control loop, the research findings can be extended to adaptive speed control of other high-load, low-speed mining machinery.

**Author Contributions:** Conceptualization, H.J.; methodology, H.J.; software, X.J., D.W. and P.L.; validation, H.J., J.C. and P.L.; formal analysis, D.W. and P.L.; resources, M.W.; data curation, Y.S.; writing—original draft preparation, H.J.; writing—review and editing, H.J.; visualization, X.J.; supervision, M.W.; project administration, H.J. and M.W.; funding acquisition, M.W. and J.C. All authors have read and agreed to the published version of the manuscript.

**Funding:** This research was funded in part by the National Natural Science Foundation of China under Grant No.52474187 and in part by the Fundamental Research Funds for the Central Universities (Ph.D. Top Innovative Talents Fund of CUMTB) under Grant No. BBJ2024062.

**Data Availability Statement:** The data used to support the findings of this study are included in the paper.

**Conflicts of Interest:** Author Penghui Li was employed by the company CCTEG Taiyuan Research Institute Co., Ltd. The remaining authors declare that the research was conducted in the absence of any commercial or financial relationships that could be construed as a potential conflict of interest.

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
