# Peer review of "Research on Speed Control Strategies for Explosion-Proof Diesel Engine Monorail Cranes"

_actuators, doi:10.3390/act13120467_

Round 1
Reviewer 1 Report
Comments and Suggestions for Authors
This manuscript addresses a challenging problem but it cannot be published because:
1) The contribution is not significant enough: it reduces only 3% the overshot at the cost of having a larger settling time (entering the band of 2% of the final value) if compared to the PID, as Figure 16 shows. This improvement is questionable.
2) How do the parameters of the PID have been tuned?. Which criterium has been used to decide the best controller by trial and error?. It seems that better controllers can be found using analytical methods.
3) In fact, introducing a well known modification of the PID law (37) (in Laplace transforms):
u(s) = -kp*\dot{\theta}_{m}(s) - kd*s*\dot{\theta}_{m}(s) + (ki/s)*e(s)
would reduce significantly the overshoot without needing any fuzzy adaptation.
4) How do the parameters of the fuzzy law have been tuned?. What are they?.
5) Errors:
- Expressions (33) and (35) are incorrect.
- In (12), is B_T or C_T?.
- Below (11), is m_v or m_x?.
6) Figure 17 is illegible.
7) Too much space of the manuscript has been devoted to detailing modelling and simulation and little space to the details of control design and performance, which is the core of the research.
Author Response
Dear revierwer:
Thank you very much for taking the time to review this manuscript (ID:actuators-3236047). Those comments are all valuable and very helpful for revising and improving our paper. We have studied comments carefully and have made correction which we hope meet with approval. Please find the detailed responses below and the corresponding revisions/corrections highlighted/in track changes in the re-submitted files.We are grateful when reading the comments.. We would like to present our minds as following:
Comments 1: The contribution is not significant enough: it reduces only 3% the overshot at the cost of having a larger settling time (entering the band of 2% of the final value) if compared to the PID, as Figure 16 shows. This improvement is questionable.
Response 1: Thank you for pointing this out. We agree with this comment. Therefore, we have readjusted the program and relevant parameters of the Fuzzy PID rules, and we have rewritten the content from line 540 to line 551 in Chapter 4.
Comments 2: How do the parameters of the PID have been tuned?. Which criterium has been used to decide the best controller by trial and error?. It seems that better controllers can be found using analytical methods.
Response 2: Thank you for pointing this out. In this study, we employ the critical proportionality method to adjust the PID parameters and derive empirical calculation results for the associated PID parameters. Subsequently, we conduct further fine-tuning based on these initial findings. The tuning process is re-edited in lines 510 to 523 of Chapter 4 of the article.
Comments 3: In fact, introducing a well known modification of the PID law (37) (in Laplace transforms): u(s) = -kp*\dot{\theta}_{m}(s) - kd*s*\dot{\theta}_{m}(s) + (ki/s)*e(s) would reduce significantly the overshoot without needing any fuzzy adaptation.
Response 3: Thank you for pointing this out. We have verified the control rate you provided. While the overshoot is effectively managed under step response conditions, the fuzzy adaptive algorithm demonstrates distinct advantages during the process of adjusting speed control targets. For a comprehensive comparison, please refer to Chapter 5 of the manuscript.
Comments 4: How do the parameters of the fuzzy law have been tuned?. What are they?.
Response 4: Thank you for pointing out this problem. We have readjusted the program and relevant parameters of the Fuzzy PID rules, and we have rewritten the content from line 540 to line 551 in Chapter 4.
Comments 5: Errors:
- Expressions (33) and (35) are incorrect.
- In (12), is B_T or C_T?.
- Below (11), is m_v or m_x?.
Response 5: Thank you for pointing out this problem. Consequently, we reviewed more specific parameters, rederived equations (33) and (35), and modified them to conform to standard format. In equation (12), the variable should be CT , while in equation (11), it should be mv. Additionally, we replaced all related parameters in the corresponding equations with their correct forms.
Comments 6: Figure 17 is illegible.
Response 6: Agree. We have, accordingly modified the plotting style of Figure 17(now Figure 18), where the dashed line with arrows highlights an enlarged view of the response curve, enabling readers to better comprehend the strengths and weaknesses of various control algorithms.
Comments 7: Too much space of the manuscript has been devoted to detailing modelling and simulation and little space to the details of control design and performance, which is the core of the research.
Response 7: Thank you for pointing out this problem. Consequently, we have included detailed explanations of the controller design in lines 510 to 519, 540 to 551, and 561 to 570 of Chapter 4. Furthermore, we have re-summarized and elaborated on the response performance of each control algorithm under various input signals in Chapter 5, with the aim of meeting your requirements.
We kindly request your review of these attachments and express our gratitude once again.

Reviewer 2 Report
Comments and Suggestions for Authors
The authors focused on optimizing speed control for a monorail crane using a fuzzy adaptive PID controller. They analyze the drive unit's structure and load characteristics, develop a mathematical model for the hydraulic speed control system, and condct simulations using AME Sim and Simulin. The fuzzy adaptive PID controller outperforms traditional PID control, what is rather expected, reducing overshoot, improving speed tracking during acceleration and deceleration, and enhancing stability. These findings highlight the controller's potential for application in other high-load, low-speed machinery, though further experimental validation is suggested.
Generally, this work is well written with attention to many details worth a book chapter rather than a short article in journal. But, after accepting the length of the contents, the following issues have not been addressed or refferred to sufficiently well. It means, a suggestion: the work has a potential of improvement or continuation by introducing real-time adaptive control like MPC for dynamic environments and data-driven optimization using machine learning for real-world validation; describe potential experimental stand that could allow to perform the optimizatio. Moreover, the authors are asked to referr to (in the text): Robust control methods like H∞ control usually enhances stability against disturbances; assessment of that application is required. Adding some energy efficiency optimization and fault detection systems would improve operational performance and reliability; assessment of that is required. Expanding to multivariable control and making the system somehow scalable to other machinery would broaden its industrial applicability; assessment of that expansionis required. Lastly, incorporating stochastic modeling would capture real-world uncertainties; assessment of further improving robustness and adaptability is required with a reference to weakness of current results.
My final recommendation follows: major revision required in the field of potential improvement of the work againts existing techniques and approaches (a section on critical assessment could be inserted).
Author Response
Dear revierwer:
Thank you very much for taking the time to review this manuscript (ID:actuators-3236047). Those comments are all valuable and very helpful for revising and improving our paper. We have studied comments carefully and have made correction which we hope meet with approval. Please find the detailed responses below and the corresponding revisions/corrections highlighted/in track changes in the re-submitted files.We would like to present our minds as following:
Comments 1: The authors focused on optimizing speed control for a monorail crane using a fuzzy adaptive PID controller. They analyze the drive unit's structure and load characteristics, develop a mathematical model for the hydraulic speed control system, and condct simulations using AME Sim and Simulin. The fuzzy adaptive PID controller outperforms traditional PID control, what is rather expected, reducing overshoot, improving speed tracking during acceleration and deceleration, and enhancing stability. These findings highlight the controller's potential for application in other high-load, low-speed machinery, though further experimental validation is suggested.
Generally, this work is well written with attention to many details worth a book chapter rather than a short article in journal. But, after accepting the length of the contents, the following issues have not been addressed or refferred to sufficiently well. It means, a suggestion: the work has a potential of improvement or continuation by introducing real-time adaptive control like MPC for dynamic environments and data-driven optimization using machine learning for real-world validation; describe potential experimental stand that could allow to perform the optimizatio. Moreover, the authors are asked to referr to (in the text): Robust control methods like H∞ control usually enhances stability against disturbances; assessment of that application is required. Adding some energy efficiency optimization and fault detection systems would improve operational performance and reliability; assessment of that is required. Expanding to multivariable control and making the system somehow scalable to other machinery would broaden its industrial applicability; assessment of that expansionis required. Lastly, incorporating stochastic modeling would capture real-world uncertainties; assessment of further improving robustness and adaptability is required with a reference to weakness of current results.
My final recommendation follows: major revision required in the field of potential improvement of the work againts existing techniques and approaches (a section on critical assessment could be inserted).
Response 1:
Thank you very much for your valuable suggestions.
We have conducted comprehensive research on your proposal to utilize the MPC controller for real-time adaptive control in dynamic environments. We examined the application methods of the MPC controller module in Simulink and, tailored to the specific context of this paper, independently designed an MPC controller module. This module was integrated into the existing co-simulation model with AMESim, and simulations were performed to analyze the dynamic responses of the MPC controller under various signal inputs. Upon validation, the MPC controller demonstrated advantages over the previously employed control algorithms. Details of the MPC controller design have been included in lines 560 to 570 of Chapter 4 of the manuscript.
Regarding your suggestion to reference robust control methods such as H∞ control and the related research on enhancing system stability against disturbances, this is indeed a focus of our future research. In this paper, we did not discuss the impact of external disturbances on the output when establishing the transfer function and simulation model; however, this does not imply that this area of research is unimportant. It will become a key focus of our subsequent studies based on the findings of this paper. Due to space limitations, it was not addressed in this manuscript.
Your suggestion to enhance operational performance and reliability by incorporating energy efficiency optimization and fault detection systems is acknowledged. Our upcoming research aims to investigate methods for optimizing energy consumption, efficiency, and power matching under various operational conditions, building upon our established modeling framework. However, as of now, no relevant findings have emerged from our research, preventing their inclusion in this paper due to time and space limitations.
In evaluating the effectiveness of the diverse control methods employed in this paper, we have enriched the comprehensive analysis and presentation in Chapter 5. This involved summarizing the impact of various control methods under varying input conditions. Additionally, we have recreated comparative graphs and tables within the text spanning from line 573 to line 609. Moreover, revisions were made to refine the abstract from line 14 to line 27 and the conclusion from line 610 to line 633 in Chapter 6 of the manuscript.
We are grateful for your suggestion to integrate stochastic modeling to address real-world uncertainties, offering fresh perspectives for our forthcoming research endeavors. We sincerely appreciate your professional and forward-thinking suggestions.
The revised manuscript has been re-uploaded, and we respectfully seek your evaluation of these materials. Thank you for your continued expert advice; we hope that our revisions align with your expectations.
Round 2
Reviewer 1 Report
Comments and Suggestions for Authors
The article has not be improved:
Previous comment 2: an heuristic procedure has been used to tune the controller. More effective methods based on, e.g. the root locus or the frequency response can be used that should yield better results.
Previous comment 3: the suggested controller, that effectively reduces overshoots, has neither been simulated nor its response shown.
An MPC controller is now proposed, It was not included in the previous version. This controller has to be described much better (its design and its control law). Moreover, it ends up that this controller performs better than the "intelligent" controller. Then the fuzzy controller can be removed and the title of the paper must be changed: "Research on Intelligent Speed Control .." does not make sense since the best controller is not an intelligent one.
Author Response
Dear revierwer:
Thank you very much for taking the time to review this manuscript (ID:actuators-3236047). Those comments are all valuable and very helpful for revising and improving our paper. We have studied comments carefully and have made correction which we hope meet with approval. Please find the detailed responses below and the corresponding revisions/corrections highlighted(The content added and modified this time is all in green font)/in track changes in the re-submitted files.
Previous comment 2: an heuristic procedure has been used to tune the controller. More effective methods based on, e.g. the root locus or the frequency response can be used that should yield better results.
Response 1: Thank you for pointing this out. The PID controller design process, refined using root locus and frequency response methods, has been expanded to Chapter 4, spanning from line 529 to line 602. Through this approach, the tuned PID controller is expected to enhance the response characteristics of the initial transfer function, achieving an overshoot of 3% and a settling time of 0.232 seconds.
Previous comment 3: the suggested controller, that effectively reduces overshoots, has neither been simulated nor its response shown.
Response 2: Thank you for pointing this out. In response to variations in the PID control rule proposed for controlling the rate, a Simulink simulation model was developed, and parameter tuning was conducted using the trial-and-error method. The resulting step response curve exhibited minimal overshoot, with detailed findings outlined in lines 603 to 640 of Chapter 4. This study focuses on a diesel engine-driven pump-controlled motor system, necessitating assumptions about key parameters of hydraulic components like pumps, motors, and control valves during mathematical model development. Real-world system behavior may deviate from derived transfer functions due to changing operational conditions, impacting specific parameters and potentially altering the transfer function itself. Consequently, assuming the inability of traditional PID tuning parameters to dynamically adjust, subsequent simulations integrating the AMESim hydraulic model encountered suboptimal outcomes compared to fuzzy PID and MPC controllers.
Another Comments: An MPC controller is now proposed, It was not included in the previous version. This controller has to be described much better (its design and its control law). Moreover, it ends up that this controller performs better than the "intelligent" controller. Then the fuzzy controller can be removed and the title of the paper must be changed: "Research on Intelligent Speed Control .." does not make sense since the best controller is not an intelligent one.
Response 3: Thank you for pointing this out. The derivation of the MPC cost function is presented between lines 684 and 720 in Chapter 4. In principle, the controller is adjusted by finding the optimal input that satisfies the cost function constraints. Since the hydraulic speed control system of the monorail crane studied in this paper may have varying system parameters under different operating conditions, the open-loop and closed-loop transfer functions derived in this paper represent ideal transfer functions under certain assumptions. However, we believe these functions are still important, as the derivation process can also guide the establishment and parameter design of the AMESim hydraulic simulation model. In the simulation study of the speed control strategy within a joint simulation environment using both Simulink and AMESim models, the controller is interfaced with the simulation model rather than with the open-loop function. This approach better reflects the impact of real operating conditions on the system. The advantage of the fuzzy PID controller lies precisely in its ability to handle systems where the accuracy of the mathematical model solution is low, the tuning parameters cannot adapt to system parameter changes, and there is significant disturbance. Therefore, the fuzzy PID-related design and research content are retained in this paper. Finally, we believe the term 'intelligent' in the title of the paper can be removed, and corresponding modifications have been made.
The revised manuscript has been re-uploaded, and we respectfully seek your evaluation of these materials. Thank you for your continued expert advice; we hope that our revisions align with your expectations.

Reviewer 2 Report
Comments and Suggestions for Authors
I have no comments to the revision.
Author Response
Dear reviewer:
Thank you for your valuable feedback on our manuscript and recognition of our revisions!
Round 3
Reviewer 1 Report
Comments and Suggestions for Authors
This manuscript still needs a lot of improvement:
1) The "intelligent controller" must be removed unless you effectively show any real advantage of it over the predictive controller. For example, you must show simulations that prove superior performance of the "intelligent controller" when process parameters change or they are not the nominal ones.
2) Too much space is devoted to the design of the PID. Just define the specifications, write the controller transfer function, plot the root locus of the controlled system, and plot the step response of the controlled system.
Author Response
Comment 1: The "intelligent controller" must be removed unless you effectively show any real advantage of it over the predictive controller. For example, you must show simulations that prove superior performance of the "intelligent controller" when process parameters change or they are not the nominal ones.
Response 1: Thank you for pointing this out. We have removed the relevant description of the intelligent controller in the new version of the manuscript.
Comment 2: Too much space is devoted to the design of the PID. Just define the specifications, write the controller transfer function, plot the root locus of the controlled system, and plot the step response of the controlled system.
Response 2: Thank you for pointing this out. We have reduced the content related to the tuning of the PID controller using the root locus method, retaining the key points you mentioned. This section can be found in Chapter 4, lines 528 to 570.
